# DNA Barcoding in Meat Authentication: Principles, Applications, and Future Perspectives

**DOI:** 10.3390/foods14203522

**Published:** 2025-10-16

**Authors:** Jiangyao Hu, Hewen Wei, Yanjie Jiang, Qingyu Xue, Feijuan Wang

**Affiliations:** 1Key Laboratory of Specialty Agri-Product Quality and Hazard Controlling Technology of Zhejiang Province, College of Life Science, China Jiliang University, Hangzhou 310018, China; hujiangyao2000@163.com (J.H.); 19700780211@163.com (Q.X.); 2Jinhua Institute of Food and Drug Inspection and Testing, Jinhua 321000, China; 13575914876@163.com (H.W.); yamajiang@126.com (Y.J.)

**Keywords:** DNA barcoding technology, meat identification, application, challenge

## Abstract

DNA barcoding technology, as a species identification method based on specific DNA sequence variations, has been widely applied in meat product authentication in recent years. This paper reviews the technical principles, current applications, and comparative advantages of DNA barcoding in meat identification, particularly in contrast to traditional authentication methods. It further highlights the critical role of DNA barcoding in ensuring meat authenticity, enhancing food safety, and contributing to biodiversity conservation efforts. Furthermore, the paper explores the strategic implications and future trends of DNA barcoding in food regulation and ecological protection, demonstrating its practical feasibility and broad prospects in meat products. By highlighting its applications in detecting food adulteration and verifying species origin, this review aims to promote the safety and sustainable development of the meat industry while providing valuable insights for related fields. Ultimately, the implementation of DNA barcoding technology serves as a crucial safeguard for public food safety and health, aligning with the growing demand for improved food control systems.

## 1. Introduction

With the substantial improvement in living standards, the demand for meat products among Chinese consumers has been demonstrating a year-on-year growth trend. However, the food industry has been plagued by alarming safety incidents in recent years. Unscrupulous businesses, driven by profit motives, have engaged in fraudulent practices, such as substituting premium meats like beef and lamb with cheaper alternatives including pork and duck [1], or even using lymph meat as pork belly. For example, at the 2024 CCTV 3·15 Gala, it was revealed that lymphatic meat was used in pre-prepared preserved mustard greens with braised pork dishes. These deceptive practices not only disrupt market integrity but also pose serious threats to consumer health and rights. Meat adulteration in animal-derived foods can also lead to religious conflicts in certain regions [2] and has become a key factor affecting food quality and safety. As international food trade expands and consumer awareness grows, the development of rapid and accurate methods for detecting meat adulteration has become an urgent necessity. This challenge directly impacts the protection of consumers’ “meat basket” [3,4,5].

As a staple in daily diets, meat products are widely cherished by consumers but have also become a hotspot for fraudulent practices [6]. Counterfeit meat products may contain harmful substances or uninspected raw materials, posing significant risks of food poisoning and disease transmission. Such practices not only endanger consumer health but also create unfair competition for legitimate producers, undermining their economic interests and discouraging compliance with legal and ethical standards. In addition, consumers who purchase adulterated meat products not only end up wasting money, but may also be subjected to direct food safety and health risks, such as potential allergic reactions. The proliferation of counterfeit meat further erodes brand reputation and consumer trust, ultimately hindering the sustainable development of the industry. Thus, addressing issues such as meat adulteration, mislabeling, and unclear sourcing is vital for industry growth, maintaining market order, and safeguarding consumer rights. Traditional detection methods, including morphological observation, protein analysis, or chemical marker identification, exhibit significant limitations when applied to processed [7], mixed, or micro-adulterated products. Therefore, traditional detection methods fail to meet the increasingly stringent requirements of both regulatory agencies and consumers, underscoring the urgent need to develop advanced, reliable, and efficient detection technologies.

In this context, DNA barcoding technology has emerged as a revolutionary tool in meat identification [8]. It uses specific DNA regions (e.g., mitochondrial or nuclear DNA) as “barcodes,” amplified via PCR and sequenced for comparison against standard databases to identify species [9]. DNA barcoding offers the significant advantage of high sensitivity, enabling the detection of even trace amounts of DNA. This makes it possible to accurately identify species in processed or heavily treated food products. Another key strength is its high accuracy—by comparing sequences against standardized reference databases, it can reliably determine the species and origin of meat samples [10]. In food safety applications, DNA barcoding can be used to test a wide range of meat products in circulation to verify whether their actual contents match the species and sources declared on the label. This helps ensure product quality and safety, effectively preventing illegal additives and adulteration, and supports the stability and fairness of market order. Consumers can also benefit from DNA barcoding by gaining transparent access to the true composition of food products—for example, by confirming whether a product labeled as beef, pork, or lamb actually contains the declared species. This reduces the risk of mislabeling or fraud, ensuring that consumers receive exactly what they expect. With the rapid adoption of high-throughput sequencing technologies, ongoing improvements in bioinformatics algorithms, and the collaborative development of global databases, DNA barcoding has transitioned from a laboratory research tool to a practical, industry-level solution. It now plays an important role in species identification [11], adulteration detection, and the protection of endangered species [12], providing strong technical support for food safety and biodiversity conservation. However, the broader implementation of DNA barcoding still faces several challenges, including the need for standardized protocols, efficient methods for analyzing mixed or complex samples, and effective cost control. Addressing these issues requires further research and practical efforts to ensure that DNA barcoding can be more widely and deeply integrated into real-world production and regulatory systems. This article clarifies the technical principles, applications, and advantages of DNA barcode technology in meat identification, highlighting its key role in ensuring meat authenticity, improving food safety, and promoting biodiversity conservation. This article also highlights DNA barcode technology’s strategic significance and future trends in food regulation and ecological protection, demonstrating its practical feasibility and broad prospects in meat products and providing an important basis for the sustainable development of the meat industry and related fields.

## 2. DNA Barcoding Technology

### 2.1. The Principle of DNA Barcoding Technology

The main DNA barcodes applied in the animal kingdom are mitochondrial DNA (mtDNA) genes and nuclear DNA (nDNA) genes. The mitochondrial genome features a simple structure, strict maternal inheritance, absence of recombination, no sequence homology with the nuclear genome, rapid evolutionary rate, multiple copies per cell, and adherence to the molecular clock theory. Consequently, it is more easily detectable than nuclear genomes, more susceptible to genetic drift, and exhibits extensive intra- and interspecific polymorphism. Owing to these characteristics, numerous mtDNA-based molecular markers are routinely employed for species identification in animals, enabling reliable distinction and identification of phylogenetically related or closely allied species, as shown in Figure 1. Mitochondrial genes used include cytochrome c oxidase I (*COI*) [13], cytochrome b (*Cyt b*) [14,15], 12S ribosomal RNA (*12S rRNA*) [16], 16S ribosomal RNA (*16S rRNA*) [17], the displacement loop region (*D-loop*) [18], and NADH dehydrogenase (ND) [19]. Additionally, nuclear genes such as simple sequence repeats (*SSRs*), short tandem repeats (*STRs*), and single-nucleotide polymorphisms (*SNPs*) are employed for species identification, including applications like individual animal identification and parentage testing, as shown in Table 1. DNA testing technology exhibits strong analytical specificity and high detection sensitivity, and its results are affected by food processing methods. It has been widely adopted for adulteration detection in meat and meat products, emerging as a critical technical approach for ensuring food safety. DNA barcodes represent specialized genomic regions, typically comprising a central hypervariable domain flanked by conserved sequences. Serving as genetic markers for organisms, they provide unique molecular signatures for species identification [20].

**Table 1 foods-14-03522-t001:** Comparison table of commonly used DNA barcode genes.

Gene Source	Gene	Characteristics	Advantages	Disadvantages	Application Scope
mtDNA	*COI*	High inter-species variation, strong discrimination; international standard barcode (iBOL recommended).	High resolution; applicable to most species; extensive database coverage.	Difficult to distinguish some closely related species; long fragment sensitive to degraded DNA.	Universal animal species identification (mammals, fish, etc.); fresh meat, conventionally processed meat products [21].
mtDNA	*Cytb*	Moderate evolutionary rate, suitable for breed-level identification; complementary to *COI* for improved accuracy.	Effective for distinguishing domesticated breeds; tolerant of moderately degraded DNA.	Lower resolution than *COI* when used alone; less comprehensive database coverage than *COI.*	Distinguishing closely related species (mammals, fish, etc.); mixed meat products, livestock breed identification [22].
mtDNA	*16S rRNA*	Highly conserved; significant inter-species variation in fish; high tolerance for degraded DNA.	Suitable for aquatic product adulteration detection; stable in complex processed samples.	Low resolution in mammals; limited universality (specific taxa only).	Aquatic product identification (fish, crustaceans); fish products, canned seafood, processed crustaceans (e.g., shrimp identification) [23].
mtDNA	*12S rRNA*	Short fragment suitable for degraded samples; highly universal.	Detects deeply processed samples (e.g., sausages, meat floss); rapid screening of low-level adulterants (<1%).	Low resolution; requires confirmation with other genes; insufficient database coverage (lacks short fragment data for some species) [24].	Highly processed meats (severely degraded DNA); cooked foods, canned meats, meat powders, and other high-temperature-processed products [25].
mtDNA	*D-loop*	Contains multiple tandem repeats and variable number tandem repeat (VNTR) regions; high mutation rate; significant individual/group variation; regulates mtDNA replication/transcription; maternal inheritance.	Rich genetic information; distinguishes closely related individuals/groups; traces maternal lineages.	Reflects only maternal inheritance; requires combined nuclear DNA analysis; rare paternal mtDNA inheritance complicates analysis; prone to mutations from internal/external factors.	Intraspecific genetic diversity studies; maternal kinship identification; preventing inbreeding to improve animal reproductive performance and offspring quality [26].
Nuclear	*SSR/* *STR*	Short tandem repeats; highly polymorphic.	High resolution for individual identification and parentage testing.	Complex allele genotyping.	Individual identification, parentage testing [27].
Nuclear	*SNP*	Single-nucleotide polymorphism; most common genomic variation.	Widely distributed; suitable for high-throughput analysis; used in association studies [27].	Requires large marker sets; high technical demands for analysis.	Gene mapping, disease-related studies [28].
Nuclear	*β-actin*	Low intraspecific polymorphism but distinguishes hybrid offspring.	Provides complementary nuclear-level information; identifies hybrids (e.g., Bos taurus × Bos indicus).	Limited resolution when used alone; requires combination with mtDNA genes.	Auxiliary identification of mammalian breeds and hybrids; breed traceability [29] (e.g., certification of specific livestock breeds).

DNA barcoding technology enables definitive identification of animal species in meat samples—such as beef, pork, or lamb—preventing mislabeling and adulteration. It can be combined with other analytical methods to determine the geographical origin, to ensure product quality and safety. For instance, Jinhua ham producers utilize DNA barcodes to deter fraudulent substitution with non-specified meats, alternative pork breeds, or meats from unauthorized regions. When applied to meat products, this technology detects discrepancies between labeled claims (species/origin) and actual market commodities, thereby combating food fraud [30]. Compared to conventional detection methods such as morphological examination and protein analysis, DNA barcoding offers advantages including high accuracy, strong universality, and high-throughput detection, enabling effective identification of meats across various processing stages. However, its application is constrained by requirements for sample quality, limitations in the scope of universal primers, relatively high costs, and the inability to detect unknown species (Figure 2).

### 2.2. Characteristics of Common DNA Barcoding Technologies

DNA barcode technology is gradually being recognized and applied in many fields, especially in some single-species identification scenarios where it is widely used. Building upon this technological foundation, multiple derivative techniques have emerged with distinct features and application-specific adaptations. Conventional PCR amplifies *COI*/*Cytb* gene fragments followed by Sanger sequencing and alignment, which is optimal for fresh or intact specimens. This mature approach achieves species-level resolution, benefits from well-established reference databases, and offers operational simplicity at low cost. However, it exhibits high failure rates with degraded samples and limited capacity to distinguish closely related species. qRT-PCR/ddPCR (real-time quantitative PCR/droplet digital PCR) is primarily used for quantitative detection of known target species, achieving species-level (quantitative) resolution at a moderate cost. It offers high sensitivity and accurate quantification but can only detect known target species, making it suitable for meat testing in specific scenarios. Mini-barcode technology enhances the detection capability for degraded DNA samples by amplifying and sequencing short fragments (100–200 bp). It is particularly suitable for detecting processed foods or decomposed samples, and while its cost is slightly higher than traditional techniques, it enables species-level identification. It boasts a high amplification success rate, but the sequences are shorter, containing less effective information, and it cannot distinguish between some species. Ultra-barcoding employs the sequencing of complete chloroplast or mitochondrial genomes. The method demonstrates high resolution for discriminating between closely related species, cryptic species, and complex taxa, with the potential to achieve population-level discrimination depending on the genomic regions analyzed. However, it has high costs and requires strong bioinformatics capabilities. Its primary advantage lies in distinguishing closely related species and hybrids. Amplicon meta-barcoding utilizes high-throughput sequencing (NGS) to amplify targeted genomic regions, enabling analysis of multi-species composition within mixed samples. This technique is particularly suitable for environmental specimens (e.g., water, soil, air) and complex admixtures, achieving community-level (multi-species) resolution with moderate-to-high cost requirements. While offering high-throughput capacity without requiring individual specimen isolation—thus allowing simultaneous detection of multiple taxa—it remains susceptible to PCR amplification bias and exhibits limited quantitative accuracy. DNA chip/microarray technology identifies specific barcode sequences based on probe hybridization. It is used for screening known target species and achieves species-level resolution at a moderate cost. This technology enables rapid, parallel detection of multiple known species. While its application in commercial-scale screening has been partially superseded by quantitative PCR (qPCR) and Next-Generation Sequencing (NGS), it remains a viable solution for implementation in specific contexts. However, it requires pre-designed probes and is incapable of detecting unknown species.

Different DNA barcoding techniques are compared in Table 2. In practical applications, the appropriate technology should be selected based on research objectives, sample type, cost budget, etc.

## 3. Application of DNA Barcoding Technology in Meat Product Authentication

DNA barcoding technology applied in meat identification primarily relies on sequence variations in specific gene regions. By selecting one or more genetic fragments that exhibit high variability (between species) yet remain conserved (within species) as barcode regions, this technology enables accurate identification of different meat species. Commonly used barcode genes display significant sequence divergence across species while maintaining relative conservation within the same species. Therefore, the species origin of meat can be determined through sequence analysis (Table 3) [37].

Wang et al. used the mitochondrial *COI* and *12S rRNA* gene markers to assess the authenticity of 33 grilled meat samples obtained through four collection channels [38]. Successful species identification was achieved for 28 samples, encompassing domestic pig, domestic cattle, zebu, sheep, mallard, domestic goose, red junglefowl, little yellow croaker, and Humboldt squid. Adulteration was detected in nine samples (32.1% of valid samples), consistently involving mislabeling where lower-value meats were substituted for higher-value ones. The adulteration rate was 100% in canteen samples, 40% in barbecue shop samples, and 0% in samples from online platforms. Ma et al. [39] developed a multiplex PCR detection method for animal-derived foods (pork, beef, lamb, chicken, and duck) using mixed primers to determine the multiplex PCR detection conditions and specificity for these five types of animal-derived foods, ultimately establishing the primer ratios and detection limits for each component. Ding et al. [40] established a non-targeted screening method for animal-derived components in meat products based on metagenomic sequencing technology. By using universal primers to amplify the *COI* gene sequence on animal mitochondrial DNA, followed by sequencing analysis and database comparison, they achieved detection of animal-derived components. The method was tested on various meat samples, yielding accurate and reliable results that could rapidly identify unknown animal-derived components, providing technical support for combating adulteration in meat products. Miao et al. [41] developed a quantitative detection method for beef and pork content using droplet digital PCR technology to address adulteration issues in meat and meat products. By determining the relationship between DNA copy numbers and meat mass, a conversion formula was derived. Experiments showed that the measurement results of this method are close to the actual values, are not affected by external species interference, can accurately detect the content of bovine and porcine components in commercially available samples, effectively identifying adulteration phenomena, and have good market application prospects. Jawla et al. [42] have designed a paper-based loop-mediated isothermal amplification–lateral flow (LAMP-LF) assay based on mitochondrial *Cytb* gene sequences, designed primers and probes, and optimized screening of paper-based matrices (cellulose/glass fiber/nitrocellulose), LAMP components, thermal programs, and probe hybridization conditions for on-site detection of water buffalo tissue. The low-cost system exhibits high specificity and can detect target DNA as low as 10 fg, with detection taking approximately 2–3 h. Kane et al. [43] conducted RT-PCR testing on beef, lamb, pork, chicken, turkey, and horse meat samples. Among 48 samples, 10 did not match the product labels, and horse meat components prohibited for sale in the U.S. market were detected in two samples. Di et al. [44] successfully detected beef, lamb, pork, and chicken using a mitochondrial *COI* gene multiplex PCR method, with a detection sensitivity of 0.001 ng. When detecting artificially adulterated pork in lamb using this method, the detection limit could be as low as 0.1 mg, and it could also be applied to species identification in leather and down products. Wang et al. [45] conducted adulteration identification on four common types of meat (beef, lamb, pork, and duck) and related meat products, using the *COI* gene as the target gene. They established DNA barcoding identification technology for these four animal-derived foods and tested 20 batches of processed meat samples. The results showed that 90% of the samples matched the ingredients listed on the product labels. One batch of beef meatballs failed to amplify due to low meat content, and another batch of beef meatballs was found to contain duck-derived components, indicating adulteration. Ai et al. [46] used a modified SDS alkaline denaturation method and salt precipitation method to extract meat mtDNA and designed specific *Cytb* primers for PCR detection, establishing a PCR method for detecting mink-derived components from raw meat. The modified SDS alkaline denaturation method yielded DNA with higher purity and yield, and the designed primers could specifically identify mink-derived components, making the method simpler and faster. Gwak et al. [47] developed a new molecular detection method using nanoplate-based digital PCR to identify pork and chicken in processed foods. The study found that targeting the mitochondrial D-ring region and *Cytb* gene exhibited high specificity for 11 animal species. After assessing DNA through 10-fold serial dilutions, the sensitivity was comparable to real-time PCR, with a detection limit of 0.1% (*w*/*w*) for pork and chicken in beef, outperforming real-time PCR’s 1% (*w*/*w*) limit. Further validation using 27 commercial meat products yielded results consistent with the labeled species information. Therefore, this double-stranded nanoparticle digital PCR detection method provides a basis for the accurate detection of pork and chicken in meat matrices and holds significant potential for application in the food industry.

**Table 3 foods-14-03522-t003:** Detection techniques for meat DNA barcodes.

Testing Content	Genes	Testing Technology	Efficacy
Cattle, sheep	*COI,* *12S rRNA*	PCR; agarose gel electrophoresis	Universal primer amplification and sequencing [38].
	*Cytb*	Semi-universal primer quintuple PCR	The lowest detection limit is 10 fg of DNA [39].
	*COI*	PCR; agarose gel electrophoresis	Universal primer amplification and metagenomic sequencing [40].
	*β-actin*	Droplet digital PCR	Quantitative detection [41].
	*COI*	PCR; real-time quantitative PCR	Universal primer amplification and sequencing; quantitative detection [43].
Pork	*COI,* *12S rRNA*	PCR; agarose gel electrophoresis	Universal primer amplification and sequencing [38].
	*Cytb*	Semi-universal primer quintuple PCR	The lowest detection limit is 10 fg of DNA [39].
	*COI*	PCR; agarose gel electrophoresis	Universal primer amplification and metagenomic sequencing [40].
	*β-actin*	Droplet digital PCR	Quantitative detection [41].
	*COI*	PCR; real-time quantitative PCR	Universal primer amplification and sequencing; quantitative detection [43].
	*COI*	Multiplex PCR	The detection limit for adulterated pork was 0.1 mg (0.05% *wt*/*wt*) [44].
	*COI*	PCR; agarose gel electrophoresis	Universal primer amplification and sequencing [45].
	*Cytb*	PCR; agarose gel electrophoresis	Distinct target bands are shown for 6.25 ng/μL DNA after amplification and electrophoresis [46].
	*Cytb,* *D-loop*	Duplex nanoplate-based digital PCR	The minimum detectable content in the mixture is 0.1% (*w*/*w*) [47].
Duck	*COI, 12S rRNA*	PCR; agarose gel electrophoresis	Universal primer amplification and sequencing [38].
	*Cytb*	Semi-universal primer quintuple PCR	The lowest detection limit is 10 fg of DNA [39].
	*COI*	PCR; agarose gel electrophoresis	Universal primer amplification and metagenomic sequencing [40].
	*β-actin*	Droplet digital PCR	Quantitative detection [41].
	*COI*	PCR; agarose gel electrophoresis	Universal primer amplification and sequencing [45].
	*Cytb*	PCR; agarose gel electrophoresis	Distinct target bands are shown for 6.25 ng/μL DNA after amplification and electrophoresis [46].
Duck	*COI,* *12S rRNA*	PCR; agarose gel electrophoresis	Universal primer amplification and sequencing [38].
	*Cytb*	Semi-universal primer quintuple PCR	The lowest detection limit is 10 fg of DNA [39].
	*COI*	PCR; agarose gel electrophoresis	Universal primer amplification and metagenomic sequencing [40].
	*β-actin*	Droplet digital PCR	Quantitative detection [41].
	*COI*	PCR; agarose gel electrophoresis	Universal primer amplification and sequencing [45].
	*Cytb*	PCR; agarose gel electrophoresis	Distinct target bands are shown for 6.25 ng/μL DNA after amplification and electrophoresis [46].
Chicken	*Cytb*	Semi-universal primer quintuple PCR	The lowest detection limit is 10 fg of DNA [39].
	*COI*	PCR; agarose gel electrophoresis	Universal primer amplification and metagenomic sequencing [40].
	*COI*	PCR; real-time quantitative PCR	Universal primer amplification and sequencing; quantitative detection [43].
	*COI*	Multiplex PCR	The detection limit for adulterated pork was 0.1 mg (0.05% *wt*/*wt*) [44].
	*Cytb*	PCR; agarose gel electrophoresis	Distinct target bands are shown for 6.25 ng/μL DNA after amplification and electrophoresis [46].
	*Cytb,* *D-loop*	Duplex nanoplate-based digital PCR	The minimum detectable content in the mixture is 0.1% (*w*/*w*) [47].
Goose, Anser cygnoides domesticus	*COI,* *12S rRNA*	PCR; agarose gel electrophoresis	Universal primer amplification and sequencing [38].
	*COI*	PCR; agarose gel electrophoresis	Universal primer amplification and metagenomic sequencing [40].
	*β-actin*	Droplet digital PCR	Quantitative detection [41].
Gallus gallus, Larimichthys polyactis, Dosidicus gigas	*COI,* *12S rRNA*	Agarose gel electrophoresis	Universal primer amplification and sequencing [38].
Horse, donkey meat; yak, rat, sparrow, ostrich, mink, camel, squab (young pigeon), cat, pheasant, pigeon, and raccoon dog meat; spicy beef granules, pig blood curd, steak; beef skewers and mutton skewers	*COI*	PCR; agarose gel electrophoresis	Universal primer amplification and metagenomic sequencing [40].
Coturnix coturnix meat, mink meat, fox meat, meat samples, processed foods, pork floss, beef floss, ham sausage, frozen beef meatballs with juicy filling, non-meat ingredients, wheat flour, soybean flour, corn flour	*β-actin*	Droplet digital PCR	Quantitative detection [41].
Buffalo (Bubalus bubalis)	*COI,* *Cytb*	Lateral flow dipstick; agarose gel electrophoresis; real-time quantitative PCR	Point-of-care testing can detect as little as 10 fg of DNA [42].
Horse	*COI*	Real-time quantitative PCR	Universal primer amplification and sequencing; quantitative detection [43].
Donkey meat, venison, mink meat	*Cytb*	Agarose gel electrophoresis	Distinct target bands are shown for 6.25 ng/μL DNA after amplification and electrophoresis [46].

DNA barcoding technology enables precise identification of animal species in meat samples—such as beef, pork, or lamb—effectively preventing species mislabeling and adulteration to ensure consumers receive accurately labeled products. Furthermore, this technology can also be combined with other analytical methods to determine the geographical origin of meat products, thereby ensuring their quality and safety. For protected designation of origin (PDO) products like Jinhua ham, DNA barcoding deters fraudulent substitution with non-specified meats, unauthorized pork breeds, or meats from non-designated regions, thereby safeguarding the reputation of premium traditional products and protecting consumers’ legal rights. Moreover, DNA barcode technology also demonstrates the following key advantages [48]: First, it exhibits high sensitivity, capable of detecting even trace amounts of DNA, allowing accurate identification of components even in heavily processed foods. Second, it provides high accuracy; by comparing sequences against extensive reference databases of known sequences, it can precisely determine the species and origin of meat samples. Third, compared to traditional detection methods, DNA barcoding technology offers rapid and efficient advantages, enabling swift sample testing and analysis [49], and is extensively applied in meat product detection as well as modern food regulation and market distribution [50]. This enables monitoring at every stage—from farming and slaughtering to processing and retail—ensuring product quality and safety. Technology facilitates timely detection and prevention of potential food safety risks while effectively curbing illegal adulteration or fraudulent substitution. Most consumers possess only a basic awareness of DNA barcoding technology, generally recognizing its utility in meat identification but lacking detailed knowledge of its core principles or analytical procedures. Although some acknowledge its role in ensuring meat quality and safety, questions remain regarding how processing, storage, and transport factors may affect detection accuracy. Nevertheless, DNA barcoding serves as a reliable tool that enables consumers to verify food composition and trace origins with greater convenience and precision. This enhanced transparency fosters increased consumer confidence in meat products, strengthens trust in the food supply chain, and supports the development of a more robust and well-regulated market environment.

## 4. Applications of DNA Barcoding Technology in Diverse Fields

### 4.1. Seafood Identification

DNA barcoding technology has been widely applied in the identification of seafood products, successfully detecting and identifying the species of seafood products sold in the market, which is beneficial for identifying adulteration issues such as discrepancies between product labels and actual contents (Table 4).

Babett et al. [51] investigated traditional octopus pies (Tielle sétoise) from Sète, France. Using *COI* gene fragment barcode technology to test 25 products, they found that 17 contained giant flying squid, with 8 (35%) being incorrectly labeled as octopus. Although local consumers showed a preference for octopus, the findings call for improved cephalopod product labeling, enhanced industry communication, and exploration of citizen science monitoring models to support informed consumer choices. Dutrudi et al. [52] used DNA barcoding technology to verify whether fish fillet product samples matched their labels. Applying the internationally recognized *COI* gene fragment for DNA barcoding, they successfully identified 54 fish fillet products collected from Thai supermarkets, achieving 98% similarity with the GenBank and BOLD databases. However, only one sample was identified at the genus level, involving 25 species and 18 genera, with only the low-eyed toothless goby being an endangered species. Using two criteria to determine label errors, the results showed that 33.33% of the samples had label errors, while the error rate for products labeled with scientific names was only 11.11%, indicating that using scientific names to label products may reduce errors and protect consumer rights. Jiang et al. [53] conducted a survey of seafood markets in South China, performing *COI* gene amplification on 478 collected samples, identifying 156 fish species, and creating a guide map. The study found that 9.6% of fish species were mislabeled, indicating commercial fraud, and 3.8% of the identified species were threatened species, providing insights for seafood market monitoring. Rosalee et al. [54] collected 35 shark products, evaluated three DNA barcoding primer sets, identified species, and tested label accuracy. The results showed that the mini-barcoding primer set had the highest species identification success rate. The study found that some products had incorrect labels or contained endangered species components. Wang et al. [55] established a DNA barcoding-based specific detection method for salmonid fish, using DNA *COI* and *16S rRNA* gene tags to test 17 commercially available salmonid fish species. The results showed that using *COI* as the primary target and *16S rRNA* as the auxiliary target could identify pure salmon products. Amy et al. [56] used DNA barcoding to analyze globally processed eel products, finding that East Asian eels were more prevalent than European eels, Japanese eels exhibited distribution differences, and endangered American eels showed increased disease prevalence. They recommended implementing a traceability system incorporating genetic methods to prevent illegal trade. Sophia et al. [57] assessed the impact of common cooking methods on DNA barcoding technology for detecting fish products and compared the detection performance of full-length barcodes and mini-barcodes. The results showed that the SH-E mini-barcode had the highest success rate (92–94%), comparable to the full-length barcode (90%), and performed better in most cooking methods. However, sequencing levels significantly decreased in canned foods. Despite the reduced sequence length, mini-barcodes remain capable of accurately identifying fish species, demonstrating the robustness of DNA barcoding as an identification tool. DNA barcoding technology has been employed in seafood markets across multiple locations, including the Philippines, India, South Africa, Brazil, Italy, and North America, to verify the authenticity of commercial seafood products. Studies from these regions have documented instances of species substitution and labeling inaccuracies, though detection rates demonstrate considerable variation depending on geographical location and product category [58]. Collectively, the findings confirm the feasibility and reliability of DNA barcoding for the detection and authentication of fish and seafood products.

### 4.2. Ecological Monitoring and Biodiversity Conservation

DNA barcoding technology is also applied in biodiversity research. By constructing a DNA barcode database for species, rapid identification and classification of species can be achieved, providing a scientific basis for biodiversity conservation and utilization (Table 4).

Zhang et al. [59] constructed a DNA barcoding database for agarwood, analyzed 57 samples using ISSR markers, and revealed high genetic diversity and population differentiation, providing evidence for the application of DNA barcoding and ISSR markers and a scientific basis for protecting existing populations. Through DNA barcoding technology, it is possible to rapidly identify and monitor invasive species for protecting agricultural and forestry production, as well as monitoring the impact of environmental pollution on ecosystems by combining DNA barcoding technology with other technologies. Daniel et al. [60] used DNA barcoding (*COI*, *18S rRNA*, *ITS*, *28S rRNA*) to detect the origin of nematode populations in soil from Australian grain-growing regions and conducted biological classification, which is crucial for biodiversity conservation. Hu et al. [61] classified, identified, and conducted DNA barcoding analysis on 76 cockroach samples collected in China. rDNA *ITS2* and mtDNA *16S rRNA* were selected as candidate genes for molecular identification. Sequence alignment and phylogenetic analysis indicated that *16S rRNA* is a better molecular identification target for cockroaches than *ITS2*, as *ITS2* sequences exhibit significant variation across different taxonomic groups, while *16S rRNA* sequences are relatively conserved. The intra- and inter-species variation in *ITS2* (2.57% vs. 5.62%) was notable, making it suitable for molecular identification of lower cockroach species. Mark and Andrew et al. [62] studied the mitochondrial *D-loop* (479 bp) of 96 Cleveland Bay horses, identifying 11 haplotypes and 27 variable positions, with four major haplotype clusters (accounting for 89% of the total sample), indicating that Cleveland Bay horses have four major maternal lineages. Debabrata et al. [63] analyzed the mitochondrial DNA regions (mitochondrial control region and *16S rRNA* region) of black tiger shrimp and found that they exhibit high genetic diversity and recent population expansion trends. The results showed that the Chennai and Blair Port populations possess the most unique haplotypes, with all populations exhibiting significant genetic variation and differentiation. Zhang et al. [64] used *D-loop* sequences to investigate the genetic diversity and population structure of 144 individuals from eight representative populations of Chinese sturgeon (*M. nipponense*) in the Songhua River, Yellow River, Yangtze River, Pearl River, Weishan Lake, Taihu Lake, Bosten Lake, and Wanquan River in northern and southern China. The results showed that the Wanquan River population had the highest haplotype diversity and nucleotide diversity, while the Bosten Lake population had the lowest. Pairwise FST values reached highly significant levels, indicating significant genetic differentiation between populations. Gene flow was abundant between Weishan Lake and Taihu Lake, as well as between the Yellow River and the Pearl River. Phylogenetic trees and network analysis showed that the Bosten Lake population exhibited haplotype clustering and low genetic diversity. Mismatch distribution analysis and Fu’s Fs and Tajima’s D tests indicated that the Chinese sturgeon populations had not undergone expansion.

**Table 4 foods-14-03522-t004:** Application of DNA barcoding in seafood and biodiversity conservation.

Application Areas	Testing Objects	Genes	Testing Technologies	References
Seafood identification	Tielle sétoise (squid vs. octopus)	*COI*	Gene meta-barcoding	[51]
	Fish fillet	*COI*	PCR; agarose gel electrophoresis	[52]
	Complete fish specimens	*COI*		[53]
	Shark products	*COI*		[54]
	Salmonid products	*COI*, *16S rRNA*		[55]
	Unagi products	*COI*	PCR (mini-barcode)	[56]
	Salmon, tuna, scad, pollock, swai, and tilapia	*COI*	PCR (both full barcoding and mini-barcoding); agarose gel electrophoresis	[57]
Ecological monitoring and biodiversity conservation	*Aquilaria sinensi*	*ITS2*, *ITS*, *psbA-trnH*, *matK*	PCR	[59]
	*Australian cereal cyst nematodes*	*COI*, *18S rRNA*, *ITS*, *28S rRNA*		[60]
	Cockroaches	*ITS2*, *16S rRNA*		[61]
	Cleveland Bay horse	*D-loop*		[62]
	*Penaeus monodon*	*D-loop*, *12SrRNA*, *16SrRNA*, *COX1*, *Cytb*, *ND1*		[63]
	*M. nipponense*	*D-loop*	PCR; agarose gel electrophoresis	[64]

## 5. Challenges Encountered in the Application of DNA Barcoding Technology

Although DNA barcoding technology has achieved remarkable success in meat authentication, it still faces several limitations and challenges. (1) The method is highly demanding in terms of sample quality; over-processing or spoilage of meat can lead to DNA degradation, compromising the accuracy and reliability of results. (2) Universal primers have a restricted scope of application; for unusual or newly discovered species, primer redesign is often necessary. Data analysis is complex, relying on specialized bioinformatics expertise and robust database support, and the technique cannot identify unknown species—organisms absent from reference databases remain unidentifiable [65]. Moreover, detection costs are relatively high, encompassing expenses for instrumentation, reagents, consumables, and professional training. Current research highlights core issues such as incomplete database coverage and the absence of unified international standards. Although platforms like BOLD Systems have cataloged more than 300,000 animal species, genetic data for livestock breeds and regionally important economic animals are still scarce. Distinguishing closely related species is difficult and often requires multi-gene assays, further increasing both cost and complexity. (3) Divergent standards among different countries and regions may hinder global harmonization and data sharing of DNA barcoding applications. (4) The high equipment costs and technical thresholds for on-site rapid testing limit the technology’s adoption in resource-limited areas. Additionally, the application of DNA barcode technology in new meat substitutes and laboratory-grown meat is also a promising direction for future research.

## 6. Conclusions and Outlooks

DNA barcoding technology offers high accuracy. By analyzing variations in specific gene sequences, it enables the precise identification of different meat species, unaffected by morphological degradation. The technology is characterized by strong universality, employing standardized barcode regions and protocols applicable to the detection of a wide range of meats and meat products. Furthermore, it possesses high-throughput capabilities, allowing for the simultaneous processing of numerous samples, thereby enhancing detection efficiency. Conceptually, it can be understood as a standardized genetic fingerprinting technique for species identification. The identification process itself is analogous to a supermarket scanner reading a conventional barcode [66]. Critically, the technology is effective for meats at various processing stages, including raw, cooked, and cured products, as DNA can be successfully extracted and analyzed from all. Consequently, DNA barcoding is a simple, rapid, reliable, and effective molecular identification technique. It serves as a valuable tool in combating food adulteration and fraud and plays a significant role in food safety regulation and biodiversity conservation [67]. For instance, in cooked meat products, DNA barcoding can detect adulteration with lower-value meat species. For biodiversity protection, by establishing species barcode databases, it enables the rapid and accurate identification of wildlife species, aiding in tracking illegal trade and protecting endangered species [68]. In combating illegal wildlife trade, DNA barcoding can determine the species origin of confiscated animal products, providing crucial scientific evidence for law enforcement and serving as a powerful tool for species identification and monitoring. In summary, this technology enables rapid, parallel detection of multiple species and is well-suited for commercial-scale screening in many fields.

The future development of DNA barcoding technology should focus on building a globally shared DNA barcoding database, integrating DNA barcoding data resources worldwide, improving the species information and genetic data in the database, and enhancing its accuracy and completeness; developing a miniaturized marking detection system suitable for processed products to enhance detection sensitivity and accuracy and expanding the application scope of DNA barcoding technology in fields such as food quality control and market regulation; and strengthening cooperation and exchange with international food safety certification bodies to promote the incorporation of DNA barcoding technology into international food safety certification standards, thereby providing stronger technical support and regulatory measures for food safety [69]. Innovation in related technologies should be strengthened to reduce the detection costs of DNA barcoding technology, simplify operational procedures, and improve detection efficiency. Further advancements in DNA barcoding technology could enable its application beyond laboratories to frontline scenarios such as customs and markets, further expanding its application domains and market reach and establishing a comprehensive regulatory network spanning the entire supply chain from production to consumption to curb industry-wide label fraud [70].

With the continuous advancement of technology, it is expected that existing challenges will be overcome and new application scenarios will be explored. For example, portable nanopore sequencers [71] (such as the Oxford Nanopore MinION) and mature CRISPR-Cas12a [72] rapid detection technologies will enable minute-level response times, making it possible to ‘input samples and output results’ within minutes, which is suitable for scenarios such as slaughterhouses and customs. At the same time, the deep integration of artificial intelligence is transforming data analysis methods. With the incorporation of blockchain technology into testing record chains, the widespread adoption of CRISPR-Cas12a on-site rapid testing devices, and the deep integration of BOLD Systems with national agricultural gene banks, DNA barcoding technology will transition from the laboratory to broader application scenarios, such as customs, slaughterhouses, and even supermarket shelves, to establish a ‘minute-level response’ meat safety protection network. New DNA technologies will become the cornerstone of global food fraud prevention systems while also serving diverse needs such as biodiversity conservation and cultural heritage research [73,74,75].

## Figures and Tables

**Figure 1 foods-14-03522-f001:**
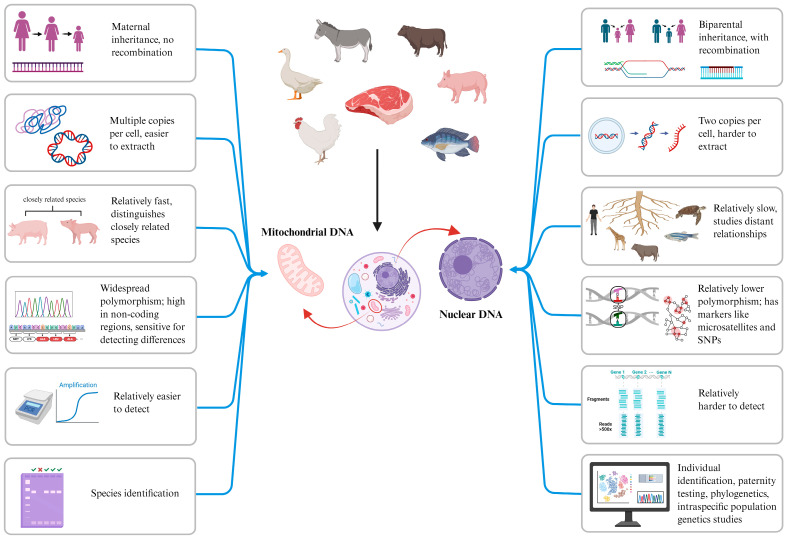
Differences between mtDNA and nDNA in species identification.

**Figure 2 foods-14-03522-f002:**
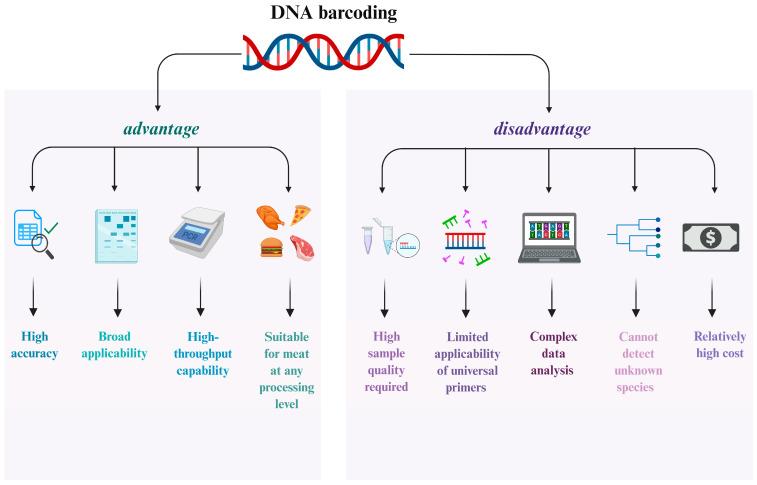
Advantages and Disadvantages of DNA Barcoding Technology.

**Table 2 foods-14-03522-t002:** Comparative analysis of DNA barcoding methodological approaches.

Category	Method	Characteristics	Application Scenarios
Traditional PCR Sequencing [31]	PCR amplification of *COI*/*Cytb* gene fragments, Sanger sequencing and comparison	High universality, well-established databases, simple operation, low cost	Fresh/intact samples (species-level)
qRT-PCR/ddPCR [32]	Real-time quantitative PCR/droplet digital PCR for quantitative detection of species-specific DNA	Rapid, sensitive, accurate quantification	Degraded samples, complex genomes (multi-species)
Mini-barcode [33]	Amplification and sequencing of short fragments (100–200 bp)	High amplification success rate, strong resistance to inhibitors	Processed or decomposed meat detection (species/subspecies-level)
Ultra-barcode [34]	Sequencing of complete chloroplast or mitochondrial genomes	Distinguishes closely related species and hybrids; extremely high resolution	Closely related species, cryptic species, complex taxonomic groups (population/multi-species)
Amplicon Meta-barcoding (AMB)	High-throughput sequencing (NGS) of amplified specific regions to analyze multi-species composition in mixed samples	High-throughput, simultaneous detection of multiple species	Environmental samples (water, soil, air), complex mixed samples (community-level, multi-species)
Metagenomic Sequencing [35]	Direct whole-genome sequencing of environmental DNA without PCR amplification	PCR-bias-free, detects unknown species, rich functional gene information	Complex mixed samples (multi-species/functional genes)
DNA Microarray/Chip [36]	Species identification via probe hybridization to specific barcode sequences	Rapid, simultaneous detection of multiple species	Commercial screening applications (multi-species)

## Data Availability

No new data were created or analyzed in this study.

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
