# Peer review of "DNA Barcoding in Meat Authentication: Principles, Applications, and Future Perspectives"

_foods, 2025, doi:10.3390/foods14203522_

Round 1
Reviewer 1 Report
Comments and Suggestions for Authors
Referee’s Report on Manuscript, “DNA Barcoding in Meat Authentication: Principles, Applications, and Future Perspectives”
General Impression: The manuscript has been well-prepared and appears to have already been formatted as per the style requirements of the journal.
There are a number of typographical errors, listed below (indicated in yellow):
Omit “The” in the first sentence, at L31.
L34: replace ‘substitute’ with ‘substituting’
L42: replace ‘rugent’ with ‘urgent’
L111: replace ‘barco des’ with ‘barcodes’
L108-109: The authors state that “DNA testing technology exhibits strong analytical specificity and high detection sensitivity, with results resistant to food processing methods.” The latter claim - regarding resistance to food processing methods - is likely regarded as contestable. Yet, further on in the manuscript, the authors also state, at L313-314 that “sequencing levels decreased significantly in canned foods” and at L371-72 that “over-processing or spoilage of meat can lead to DNA degradation, compromising the accuracy and reliability of results”. Hence, the authors may need to re-consider the specific claim at L108-109 or to otherwise amend the wording to cohere with the latter points, in this reviewer's opinion.
In Table 1, table legibility could very much benefit from increased spacing between the first and second rows, and between the second and third rows, i.e., these lines could be made more legible by increasing the separation between them.
Author Response
|
Comments 1: Omit “The” in the first sentence, at L31. |
|
Response 1: We think this is an excellent suggestion. It has been deleted. It can be found in Page 1, Line 31.
|
|
Comments 2: L34: replace ‘substitute’ with ‘substituting’ |
|
Response 2: Thank you for pointing this out. “substitute” has been revised with “substituting” in the revised manuscript. The modifications were marked in red. It can be found in Page 1, Line 35.
|
|
Comments 3: L42: replace ‘rugent’ with ‘urgent’ |
|
Response 3: Thank you for pointing this out. “rugent” has been revised with “urgent” in the revised manuscript. The modifications were marked in red. It can be found in Page 2, Line 44.
|
|
Comments 4: L111: replace ‘barco des’ with ‘barcodes’ |
|
Response 4: Thank you for pointing this out. “barco des” has been revised with “barcodes” in the revised manuscript. The modifications were marked in red. It can be found in Page 3, Line 119.
|
|
Comments 5: L108-109: The authors state that “DNA testing technology exhibits strong analytical specificity and high detection sensitivity, with results resistant to food processing methods.” The latter claim - regarding resistance to food processing methods - is likely regarded as contestable. Yet, further on in the manuscript, the authors also state, at L313-314 that “sequencing levels decreased significantly in canned foods” and at L371-72 that “over-processing or spoilage of meat can lead to DNA degradation, compromising the accuracy and reliability of results”. Hence, the authors may need to re-consider the specific claim at L108-109 or to otherwise amend the wording to cohere with the latter points, in this reviewer's opinion. |
|
Response 5: Thank you for pointing this out. “DNA testing technology exhibits strong analytical specificity and high detection sensitivity, with results resistant to food processing methods.” has been revised with “DNA testing technology exhibits strong analytical specificity and high detection sensitivity, and its results are affected by food processing methods.” . It can be found in Page 4, Line 114-116.
|
|
Comments 6: In Table 1, table legibility could very much benefit from increased spacing between the first and second rows, and between the second and third rows, i.e., these lines could be made more legible by increasing the separation between them. |
|
Response 6: Thank you for pointing this out. Table 1 has been revised in the manuscript. It can be found in Page 4-6, Line 125. |

Reviewer 2 Report
Comments and Suggestions for Authors
The authors have thoroughly compiled information on the technical principles, current applications, and comparative advantages of DNA barcoding in meat identification, especially compared to traditional authentication methods. This comprehensive review will serve as an excellent resource on the topic. After carefully reading the manuscript, I have a few observations.
- I recommend proofreading the text for typos and errors.
- The introduction should end with the aim of this work clearly mentioned.
- Phrases like "pose serious threats to consumer health and rights" and "the fundamental rights to dietary health and safety" (lines 36 and 42) can be merged. Keeping just one will maintain clarity without losing meaning.
- Lines 39 and 41: The statements “the detection of such adulteration has become a global priority” and “the development of rapid and accurate methods for detecting meat adulteration has become an urgent necessity” convey the same urgency about detecting adulteration. Keep only one to reduce repetition.
- Methodology: Please specify which databases were used. How were the references for this review chosen? What criteria determined the inclusion and exclusion of references?
- Table 3: Add the efficacy of the used techniques for each type of meat.
- Lines 265-268: Expand on the discussion on the consumer's perception and understanding of DNA barcoding technology in meat authentication.
- The references appear in two styles in the text (see Table 4). Please ensure that you follow the journal instructions.
- The subsection headings are numbered incorrectly. Please fix them.
- Add a conclusion section to provide a more concise summary of the key takeaways.
There are some typos and mistakes.
Author Response
|
Comments 1: I recommend proofreading the text for typos and errors. |
|
Response 1: Thank you for pointing this out. We have carefully proofread the manuscript. |
|
|
|
Comments 2: The introduction should end with the aim of this work clearly mentioned. |
|
Response 2: We think this is an excellent suggestion. The aim has been added at the end of the introduction. It can be found in Page 2-3, Line 90-97. Such as “This article clarifies the technical principles, applications, and advantages of DNA barcode technology in meat identification, highlighting its key role in ensuring meat authenticity, improving food safety, and promoting biodiversity conservation. It also highlights its strategic significance and future trends in food regulation and ecological protection, demonstrating its practical feasibility and broad prospects in meat products, and providing important basis for the sustainable development of the meat industry and related fields.”
|
|
Comments 3: Phrases like "pose serious threats to consumer health and rights" and "the fundamental rights to dietary health and safety" (lines 36 and 42) can be merged. Keeping just one will maintain clarity without losing meaning. |
|
Response 3: Thank you for pointing this out. They have been merged in the manuscript. It can be found in Page 1, Line 39. |
|
Comments 4: Lines 39 and 41: The statements “the detection of such adulteration has become a global priority” and “the development of rapid and accurate methods for detecting meat adulteration has become an urgent necessity” convey the same urgency about detecting adulteration. Keep only one to reduce repetition. |
|
Response 4: Thank you for pointing this out. They have been merged in the manuscript. It can be found in Page 1-2, Line 42-44. Such as “As international food trade expands and consumer awareness grows, the development of rapid and accurate methods for detecting meat adulteration has become an urgent necessity.”
|
|
Comments 5: Methodology: Please specify which databases were used. |
|
Response 5: Thank you for pointing this out. The adopted database “BOLD Systems” has been added to the manuscript. It can be found in Page 14, Line 397.
|
|
Comments 6: How were the references for this review chosen? What criteria determined the inclusion and exclusion of references? |
|
Response 6: We think this is an excellent suggestion. The keywords such as “DNA Barcoding”, ” Meat”, “Authentication” to search in the data.
|
|
Comments 7: Table 3: Add the efficacy of the used techniques for each type of meat. |
|
Response 7: We think this is an excellent suggestion. The efficacy of the used techniques has been added to table 3.
|
|
Comments 8: Lines 265-268: Expand on the discussion on the consumer's perception and understanding of DNA barcoding technology in meat authentication. |
|
Response 8: Thank you for pointing this out. The discussion has been added in Page 11, Line 278-287. Such as “Most consumers possess only a basic awareness of DNA barcoding technology, generally recognizing its utility in meat identification but lacking detailed knowledge of its core principles or analytical procedures. Although some acknowledge its role in ensuring meat quality and safety, questions remain regarding how processing, storage, and transport factors may affect detection accuracy. Nevertheless, DNA barcoding serves as a reliable tool that enables consumers to verify food composition and trace origins with greater convenience and precision. This enhanced transparency fosters increased consumer confidence in meat products, strengthens trust in the food supply chain, and supports the development of a more robust and well-regulated market environment. “
|
|
Comments 9: The references appear in two styles in the text (see Table 4). Please ensure that you follow the journal instructions. |
|
Response 9: Sorry. We have modified the styles of references in Table 4 to be consistent with those in the text.
|
|
Comments 10: The subsection headings are numbered incorrectly. Please fix them. |
|
Response 10: Thank you for pointing this out. The subtitle numbers have been revised.
|
|
Comments 11: Add a conclusion section to provide a more concise summary of the key takeaways. |
|
Response 11: Thank you for pointing this out. “6. Outlook” has been revised with ”6. Conclusions and Outlooks”, and a conclusion section has been added in the part of “6. Conclusions and Outlooks”. It can be found in Page 15, Line 428-430. Such as ”In summary, this technology enables rapid, parallel detection of multiple species and is well-suited for commercial-scale screening in many fields.” |

Reviewer 3 Report
Comments and Suggestions for Authors
The manuscript provides a broad review of DNA barcoding in meat authentication, covering its principles, applications, and future perspectives. The topic is timely and relevant, but several statements are overstated, speculative, or not sufficiently substantiated by evidence. Below are specific comments to improve clarity, justification, and alignment with the review’s stated objectives.
• Line 31–32: “…consumer demand and quality expectations for meat products have risen sharply.” Is overgeneralized. Please support with consumer data or qualify by region.
• Line 35–36: “…or even using lymph meat as pork belly.” Is anecdotal without prevalence data; risks sounding sensational. Consider citing a documented case.
• Line 45–47: “Counterfeit meat products often contain harmful substances or uninspected raw materials…”
“Often” is too strong without evidence. Please soften or provide data.
• Line 51–52: “…may result in adverse health.” is wague wording. Clarify specific risks or avoid speculative phrasing.
• Line 57–59: “These shortcomings fail to meet the increasingly stringent demands…”
This is sweeping claim. Traditional methods vary in effectiveness; refine the statement.
• Line 69–70: “Compared with traditional methods, DNA barcoding is faster and more efficient…”
This is context-dependent. Barcoding can be slower or costlier than rapid assays. Please qualify.
• Line 82–84: “…plays an irreplaceable role in…full-chain traceability…” is overstated. DNA barcoding identifies species but does not provide full traceability without integration with other systems.
• Line 121–122 / 245–248: “…verifies geographical origins…”
This is misleading. DNA barcoding confirms species identity, but provenance requires complementary methods.
• Line 128–129: “…comparative strengths and limitations detailed in Figure 2.”
This is insufficient explanation. Please summarize in the text.
• Line 134–135: “…characterized by universal applicability and high accuracy…” is overgeneralized. DNA barcoding has known limitations (e.g., hybrids, degraded DNA).
• Line 146–149: “…cost is slightly higher…some targets.”
This is vague. Provide specifics or references.
• Line 152–154: “…extremely high resolution…population/variety levels.”
This is overstated. Full mitogenomes improve resolution but not always at variety level.
• Line 163–167: “…well-suited for commercial-scale screening.”
This might be misleading. Microarrays are less common today than qPCR/NGS. Update context.
• Line 252–259: “…advantages that are difficult for other methods to match…”
This is too absolute. Other molecular methods can also be rapid and sensitive.
• Line 260–267: “…comprehensive traceability across the entire food supply chain…”
This seems illogical. Traceability requires integration with logistics and labeling systems.
• Line 277–279: “…flavour of the pie was unrelated to cephalopod species…” is tangential. Consider condensing; not directly linked to barcoding.
• Line 280–281: “…consumers indicated they were willing to make purchasing decisions based on carbon footprint and sustainability.”
This is weakly connected. Interesting but not substantiated by barcoding results.
• Line 316–319: “…successfully applied… prevalent issues such as adulteration…”
This is overgeneralized. Error rates vary by region/product; qualify the claim.
• Line 329–331: “…possible to monitor the impact of environmental pollution…”
This is overstated. Barcoding detects species but not direct pollution effects.
• Line 345–346: “…closely related to Nordic, Iberian, and North African horse breeds…”
This appears out of scope for a meat-authentication review. Consider shortening.
• Line 348–364: Detailed shrimp and sturgeon haplotype results.
This is excessive biodiversity detail; tangential to the main scope. Suggest moving to supplementary or condensing.
• Line 386–387: “…applicability to novel meat substitutes and lab-cultured meat…”
This is speculative. Please frame as a potential future direction rather than a current limitation.
Author Response
|
3. Point-by-point response to Comments and Suggestions for Authors |
|
Comments 1: Line 31–32: “…consumer demand and quality expectations for meat products have risen sharply.” Is overgeneralized. Please support with consumer data or qualify by region. |
|
Response 1: Thank you for pointing this out. We have modified with “the demand for meat products among Chinese consumers has been demonstrating a year-on-year growth trend. “ in the manuscript. It can be found in Page 1, Line 31-33.
|
|
Comments 2: Line 35–36: “…or even using lymph meat as pork belly.” Is anecdotal without prevalence data; risks sounding sensational. Consider citing a documented case. |
|
Response 2: Thank you for pointing this out. We have cited a related report “For example, at the 2024 CCTV 3·15 Gala, it was revealed that lymphatic meat was used in pre-prepared preserved mustard greens with braised pork dishes “ in the manuscript. It can be found in Page 1, Line 36-38.
|
|
Comments 3: Line 45–47: “Counterfeit meat products often contain harmful substances or uninspected raw materials…” “Often” is too strong without evidence. Please soften or provide data. |
|
Response 3: Thank you for pointing this out. “often” has been revised with “may”, such as “Counterfeit meat products may contain harmful substances or uninspected raw materials, posing significant risks of food poisoning and disease transmission. “. It can be found in Page 2, Line 47-49.
|
|
Comments 4: Line 51–52: “…may result in adverse health.” is wague wording. Clarify specific risks or avoid speculative phrasing. |
|
Response 4: Thank you for pointing this out. It has been revised in the manuscript, such as “In addition, consumers who purchase adulterated meat products not only end up wasting money, but may also pose direct food safety and health risks, such as potential allergic reactions. “ It can be found in Page 2, Line 52-54.
|
|
Comments 5: Line 57–59: “These shortcomings fail to meet the increasingly stringent demands…” This is sweeping claim. Traditional methods vary in effectiveness; refine the statement. |
|
Response 5: Thank you for pointing this out. The statement has been revised. Such as “Therefore, traditional detection methods fail to meet the increasingly stringent requirements of both regulatory agencies and consumers, underscoring the urgent need to develop advanced, reliable, and efficient detection technologies. “. It can be found in Page 2, Line 61-63.
|
|
Comments 6: Line 69–70: “Compared with traditional methods, DNA barcoding is faster and more efficient…” This is context-dependent. Barcoding can be slower or costlier than rapid assays. Please qualify. |
|
Response 6: Thank you for pointing this out. “Compared with traditional methods, DNA barcoding is faster and more efficient…”in the manuscript has been deleted.
|
|
Comments 7: Line 82–84: “…plays an irreplaceable role in…full-chain traceability…” is overstated. DNA barcoding identifies species but does not provide full traceability without integration with other systems. |
|
Response 7: Thank you for pointing this out. We have modified with “DNA barcoding has transitioned from a laboratory research tool to a practical, industry-level solution. It now plays an important role in species identification [11], adulteration detection, and the protection of endangered species [12], providing strong technical support for food safety and biodiversity conservation. “ in the manuscript. It can be found in Page 2, Line 82-85.
|
|
Comments 8: Line 121–122 / 245–248: “…verifies geographical origins…” This is misleading. DNA barcoding confirms species identity, but provenance requires complementary methods. |
|
Response 8: Thank you for pointing this out. We have modified with “It can be combined with other analytical methods to determine the geographical origin,to ensure product quality and safety. “ in the manuscript. It can be found in Page 6, Line 126-128. We have modified “this technology can also be combined with other analytical methods to determine the geographical origin of meat products, thereby ensuring their quality and safety.“ in the manuscript. It can be found in Page 11, Line 260-262. |
|
Comments 9: Line 128–129: “…comparative strengths and limitations detailed in Figure 2.” This is insufficient explanation. Please summarize in the text. |
|
Response 9: Thank you for pointing this out. We have modified with “Compared to conventional detection methods such as morphological examination and protein analysis, DNA barcoding offers advantages including high accuracy, strong universality, and high-throughput detection, enabling effective identification of meats across various processing stages. However, its application is constrained by requirements for sample quality, limitations in the scope of universal primers, relatively high costs, and the inability to detect unknown species (Figure 2). “ in the manuscript. It can be found in Page 6, Line 132-138.
|
|
Comments 10: Line 134–135: “…characterized by universal applicability and high accuracy…” is overgeneralized. DNA barcoding has known limitations (e.g., hybrids, degraded DNA). |
|
Response 10: Thank you for pointing this out. “…characterized by universal applicability and high accuracy…”has been revised with “DNA barcode technology is gradually being recognized and applied in many fields, especially in some single species identification scenarios where it is widely used”. It can be found in Page 6, Line 142-143.
|
|
Comments 11: Line 146–149: “…cost is slightly higher…some targets.” This is vague. Provide specifics or references. |
|
Response 11: Thank you for pointing this out. We have modified with “It is particularly suitable for detecting processed foods or decomposed samples, and while its cost is slightly higher than traditional techniques, it enables species-level identification.“ in the manuscript. It can be found in Page 6-7, Line 156-158.
|
|
Comments 12: Line 152–154: “…extremely high resolution…population/variety levels.” This is overstated. Full mitogenomes improve resolution but not always at variety level. |
|
Response 12: Thank you for pointing this out. We have modified with “The method demonstrates high resolution for discriminating between closely related species, cryptic species, and complex taxa, with the potential to achieve population-level discrimination depending on the genomic regions analyzed. “ in the manuscript. It can be found in Page 7, Line 162-1764.
|
|
Comments 13: Line 163–167: “…well-suited for commercial-scale screening.” This might be misleading. Microarrays are less common today than qPCR/NGS. Update context. |
|
Response 13: Thank you for pointing this out. We have modified with “This technology enables rapid, parallel detection of multiple known species. While its application in commercial-scale screening has been partially superseded by quantitative PCR (qPCR) and Next-Generation Sequencing (NGS), it remains a viable solution for implementation in specific contexts.“ in the manuscript. It can be found in Page 7, Line 176-179.
|
|
Comments 14: Line 252–259: “…advantages that are difficult for other methods to match…” This is too absolute. Other molecular methods can also be rapid and sensitive. |
|
Response 14: Thank you for pointing this out. We have modified with “Moreover, DNA barcodes also demonstrate the following key advantages[48].“ in the manuscript. It can be found in Page 11, Line 266-267.
|
|
Comments 15: Line 260–267: “…comprehensive traceability across the entire food supply chain…” This seems illogical. Traceability requires integration with logistics and labeling systems. |
|
Response 15: Thank you for pointing this out. We have modified with “Third, compared to traditional detection methods, DNA barcoding technology offers rapid and efficient advantages, enabling swift sample testing and analysis[49], and is extensively applied in meat product detection as well as modern food regulation and market distribution [50]. “ in the manuscript. It can be found in Page 11, Line 271-275.
|
|
Comments 16: Line 277–279: “…flavour of the pie was unrelated to cephalopod species…” is tangential. Consider condensing; not directly linked to barcoding. |
|
Response 16: Thank you for pointing this out. It has been deleted in the manuscript.
|
|
Comments 17: Line 280–281: “…consumers indicated they were willing to make purchasing decisions based on carbon footprint and sustainability.” This is weakly connected. Interesting but not substantiated by barcoding results. |
|
Response 17: Thank you for pointing this out. We have modified with “Although local consumers showed a preference for octopus, the findings call for improved cephalopod product labelling, enhanced industry communication, and exploration of citizen science monitoring models to support informed consumer choices.“ in the manuscript. It can be found in Page 12, Line 297-299.
|
|
Comments 18: Line 316–319: “…successfully applied… prevalent issues such as adulteration…” This is overgeneralized. Error rates vary by region/product; qualify the claim. |
|
Response 18: Thank you for pointing this out. We have modified with “DNA barcoding technology has been employed in seafood markets across multiple countries including the Philippines, India, South Africa, Brazil, Italy and North America to verify the authenticity of commercial seafood products. Studies from these regions have documented instances of species substitution and labelling inaccuracies, though detection rates demonstrate considerable variation depending on geographical location and product category “ in the manuscript. It can be found in Page 13, Line 334-338.
|
|
Comments 19: Line 329–331: “…possible to monitor the impact of environmental pollution…” This is overstated. Barcoding detects species but not direct pollution effects. |
|
Response 19: Thank you for pointing this out. We have modified with “Through DNA barcoding technology, it is possible to rapidly identify and monitor invasive species for protecting agricultural and forestry production, as well as monitoring the impact of environmental pollution on ecosystems by combining with other technologies. “ in the manuscript. It can be found in Page 13, Line 349-351.
|
|
Comments 20: Line 345–346: “…closely related to Nordic, Iberian, and North African horse breeds…” This appears out of scope for a meat-authentication review. Consider shortening. |
|
Response 20: Thank you for pointing this out. The sentence has been deleted in the manuscript.
|
|
Comments 21: Line 348–364: Detailed shrimp and sturgeon haplotype results. This is excessive biodiversity detail; tangential to the main scope. Suggest moving to supplementary or condensing. |
|
Response 21: Thank you for pointing this out. But we still think it should be retained, which is also the application of DNA barcode technology in detecting species genetic diversity.
|
|
Comments 22: Line 386–387: “…applicability to novel meat substitutes and lab-cultured meat…” This is speculative. Please frame as a potential future direction rather than a current limitation. |
|
Response 22: Thank you for pointing this out. We have modified with “Additionally, the application of DNA barcode technology in new meat substitutes and laboratory grown meat is also a promising direction for future research. “ in the manuscript. It can be found in Page 14, Line 404-406. |

Round 2
Reviewer 2 Report
Comments and Suggestions for Authors
The revised manuscript demonstrates significant improvement following the revisions. The authors have thoughtfully addressed all of the suggested comments, enhancing the clarity and overall quality of the submission.